# Effects of Postbiotics and Paraprobiotics as Replacements for Antibiotics on Growth Performance, Carcass Characteristics, Small Intestine Histomorphology, Immune Status and Hepatic Growth Gene Expression in Broiler Chickens

**DOI:** 10.3390/ani12070917

**Published:** 2022-04-03

**Authors:** Yohanna Danladi, Teck Chwen Loh, Hooi Ling Foo, Henny Akit, Nur Aida Md Tamrin, Mohammad Naeem Azizi

**Affiliations:** 1Department of Animal Science, Faculty of Agriculture, Universiti Putra Malaysia (UPM), Serdang 43400, Selangor, Malaysia; avyohanna@gmail.com (Y.D.); henny@upm.edu.my (H.A.); nuraida.mtamrin@upm.edu.my (N.A.M.T.); naimazizi83@gmail.com (M.N.A.); 2Institute of Tropical Agriculture and Food Security, Universiti Putra Malaysia (UPM), Serdang 43400, Selangor, Malaysia; 3Department of Bioprocess Technology, Faculty of Biotechnology and Biomolecular Sciences, Universiti Putra Malaysia (UPM), Serdang 43400, Selangor, Malaysia; 4Institute of Bioscience, Universiti Putra Malaysia (UPM), Serdang 43400, Selangor, Malaysia

**Keywords:** antibiotics, postbiotics, paraprobiotics, growth performance, carcass, histomorphology, immune status, broiler chickens

## Abstract

**Simple Summary:**

This study aimed to determine the effects of replacing antibiotics with postbiotics and paraprobiotics on growth performance, small intestine morphology, immune status, and hepatic growth gene expression in broiler chickens. Different strains of postbiotics and paraprobiotics were prepared from the active culture of *Lactiplantibacillus plantarum* and incorporated into the broiler starter and finisher diets at a 0.2% level. Birds were fed with the formulated diets for a period of 35 days. Bodyweight (BW) and feed intake (FI) were measured weekly, and the data were used to calculate body weight gain (BWG) and feed conversion ratio (FCR). The European Broiler Index (EBI) was used to compare the broiler results from different treatments. The EBI is a factor that standardizes technical results by taking into account feed conversion, mortality, and daily gain. EBI was calculated using the formula: (Average grams gained/day × % survival rate)/Feed conversion × 10. At the end of the feeding trial, birds were euthanized, and samples were collected for small intestine morphology, immune status, and hepatic growth gene expression determination. The results revealed that growth performance parameters were not affected by the dietary treatments. However, feed intake was significantly affected both at the starter and finisher phases. The abdominal fat, colon mucosa sIgA, histomorphology, and growth gene expression were significantly affected by the treatment diets. IgM was only significantly different at the finisher phase.

**Abstract:**

Background: This experiment was designed to investigate how replacing antibiotics with postbiotics and paraprobiotics could affect growth performance, small intestine morphology, immune status, and hepatic growth gene expression in broiler chickens. Methods: The experiment followed a completely randomized design (CRD) in which eight treatments were replicated six times with seven birds per replicate. A total of 336, one-day-old (COBB 500) chicks were fed with the eight treatment diets, which include T1 = negative control (Basal diet), T2 = positive control (Basal diet + 0.01% (*w*/*w*) Oxytetracycline), T3 = Basal diet + 0.2% (*v*/*w*) postbiotic TL1, T4 = Basal diet + 0.2% (*v*/*w*) postbiotic RS5, T5 = Basal diet + 0.2% (*v*/*w*) paraprobiotic RG11, T6 = Basal diet + 0.2% (*v*/*w*) postbiotic RI11, T7 = Basal diet + 0.2% (*v*/*w*) paraprobiotic RG14, T8 = Basal diet + 0.2% (*v*/*w*) paraprobiotic RI11, for 35 days in a closed house system. Results: The growth performance indicators (final body weight, cumulative weight gain, and feed conversion ratio) were not significantly (*p* > 0.05) affected by the dietary treatments. However, feed intake recorded a significant (*p* < 0.05) change in the starter and finisher phases across the dietary treatments. Paraprobiotic RG14 had significantly (*p* < 0.05) lower abdominal fat and intestines. Villi heights were significantly (*p* < 0.05) increased, while the crypt depth decreased significantly due to dietary treatments. The dietary treatments significantly influenced colon mucosa sIgA (*p* < 0.05). Similarly, plasma immunoglobulin IgM level recorded significant (*p* < 0.05) changes at the finisher phase. In this current study, the hepatic GHR and IGF-1 expressions were significantly (*p* < 0.05) increased by postbiotics and paraprobiotics supplementation. Conclusions: Therefore, it was concluded that postbiotics and paraprobiotics differ in their effect on broiler chickens. However, they can replace antibiotics without compromising the growth performance, carcass yield, and immune status of broiler chickens.

## 1. Introduction

Antibiotics as a growth promoter have brought about high achievements in poultry production output worldwide [1]. Antibiotics usage in animal feeds has brought about the change of intestinal flora in chickens and influenced the immunity of the chickens with an increase in capacity to control diseases [1,2]. However, the uncontrolled and indiscriminate usage of antibiotics with those resultant antibiotic-resistant bacteria and increased incidence of antibiotics residues in animal products has caused harmful effects on the health of animals and consumers [3,4]. Therefore, a ban was imposed on antibiotics usage in animal feeds among the European countries and most countries of the world. Hence, the search for safer replacements with the same or better effects on animal production than antibiotics has been vigorously sought [5].

Thus, several antibiotic substitutes in livestock production such as prebiotics, probiotics, symbiotics, and postbiotics were extensively investigated in recent times [6,7,8]. A probiotic has been defined as a viable microorganism, a sufficient amount of which reaches the intestine in an active state and thus exerts positive health effects [9]. Probiotics are a natural microbial population with antimicrobial activity [10]. Probiotics can also be defined as direct-fed microbial (DFM) that could be single or a mixture of the culture of living non-pathogenic microorganisms, which administered confers health benefits on the host [11]. A postbiotic is any factor resulting from the activity of a probiotic or any released molecule capable of conferring beneficial effects to the host directly or indirectly [9]. Postbiotics are non-viable bacterial products or metabolic by-products from probiotic microorganisms with biological activity in the host [9,12].

Probiotics, for example, colonize the GIT, increase the natural microbial environment, and hinder the increase in disease-causing organisms [7]. The results from molecular and genetic studies revealed mechanisms of probiotics that demonstrate positive effects on the host. These mechanisms include immunomodulation of the host, inhibiting bacteria toxins, production antagonism through the production of antimicrobial substances, and competition with pathogens for adhesion to the epithelium and nutrients [13,14,15]. Despite the numerous benefits of probiotics, several genes contained in the cells are resistant to antibiotics that may easily cross between organisms [16]. Therefore, with time, live bacteria probiotics usage could be discouraged. 

The most common bacterial species used as probiotics belong to the LAB family, including *L. plantarum*, *L. bulgaricus*, *L. acidophilus*, *L. casei*, *L. lactis*, *E. faecalis*, *Bifidobacterium* sp. [17]. Generally, the health benefits provided by probiotic microbes are classified into three levels according to their site of action. The levels include having direct interaction with the microbiota within the gut lumen and can also have a direct metabolic effect in the gut by providing enzymatic activities; interacting with the gut mucus and the epithelium, including barrier effects, digestive processes, mucosal immune system, and enteric nervous system; and transmits signals to the host beyond the gut to the liver, systemic immune system and other important organs such as the brain [18]. Some mechanisms of action of probiotics have been reported. They include competitive exclusion, promoting gut maturation and integrity, regulating the immune system, preventing inflammation, improving growth, providing metabolism, improving the fatty acid profile, and oxidative stability in fresh meat [19].

Despite the numerous health benefits that probiotics provide to host animals in combating diseases, there are some problems in feeding live (viable) probiotic cells. Notably among the problems is how dependent the viability of microbes is on certain storage requirements; many probiotic bacteria lose their desired viability during storage [20]. Probiotic microbes are host-specific in their ability to colonize and persist in the GIT of the host. Therefore, getting a suitable strain of probiotics suitable to the host becomes difficult to achieve in practical terms [21,22]. Another important issue to note is the timing of their application, which is important in the colonization of microbes, and this colonization is temporary [23,24,25]. Furthermore, the possibility of horizontal transfer of virulent genes from pathogenic microbes to probiotic bacteria in the host is very high [16,26].

Recently, vast attention has been shifted to metabolic by-products of probiotics, known as “postbiotics”, which are preferred substitutes for probiotics. The research findings showed that postbiotics demonstrated action like probiotics [27,28,29]. Postbiotics possess probiotic effects without the living cells [28,29,30]. Therefore, postbiotics possessed most of the benefits of probiotics virtually. Improvement of gut health was a major benefit of postbiotics to animal nutrition. Other benefits include inhibition of harmful bacteria growth resulting in proper nutrient utilization and increased growth of animals [31,32,33]. Another novel product most recently discovered was the non-viable (dead), non-culturable, and possibly immunologically active cell (paraprobiotic) revealed to confer health benefits to the host and has received wide attention [22]. The term paraprobiotics was coined to indicate the use of inactivated microbial cells (non-viable) or cell fractions to confer health benefits to consumers [34]. Paraprobiotics were also defined as “inactivated probiotics” and ghost probiotics [12]. Paraprobiotics produced from microorganisms will generally lose their viability after exposure to certain factors. These factors can cause alteration in microbial cell structures, including breaking of DNA filaments, disruption of the cell membrane, or mechanical damage to the cell envelope. Furthermore, it causes changes in microbial physiological functions such as the inactivation of key enzymes or deactivation of membrane selectivity [35,36]. Several benefits provided to the host by paraprobiotics include modulation of the immune system (compounds of the cell wall may boost the immunological system) [37,38,39]. It also assists in increasing adhesion to the intestinal cell, which further inhibits pathogens [40]. The metabolites secreted from the cell was also reported to have health benefits [41].

*Lactiplantibacillus plantarum*, which was previously known as *Lactobacillus plantarum* [42], is a member of LAB, which has been popular in livestock nutrition and regarded as “Generally Recognized As Safe” (GRAS) status [9]. Several reports indicate that postbiotics produced from *L. plantarum* have exhibited broad antagonistic activities, demonstrating their potential to inhibit pathogens of various species [32,33,43]. Furthermore, postbiotics have caused improvement in broilers, laying hens, and pigs in terms of growth, meat quality, fecal lactic acid bacteria, villus height, and ability to withstand heat stress [27,28,29,30,44]. However, the effect of *L. plantarum* paraprobiotics on growth performance, carcass yield, small intestines histomorphology, immune status, and hepatic growth gene expression in broilers has not been fully studied. Therefore, this study was aimed to investigate the effects of postbiotics and paraprobiotics as replacements for antibiotics on growth performance, small intestines histomorphology, immune status, and hepatic growth gene expression.

## 2. Materials and Methods

### 2.1. Postbiotic and Paraprobiotic Preparations

The postbiotics and paraprobiotics were prepared from the following strains of *L. plantarum* (RG11, RG14, RI11, RS5, and TL1). The active culture of *L. plantarum* strains was washed once with sterile 0.85% (*w*/*v*) NaCl (Merk, Darmstadt, Germany) solution and adjusted to 10^9^ CFU/mL to be used as a 10% (*v*/*v*) inoculum according to the method described by Mohamad et al. [45]. Both postbiotics and paraprobiotics were prepared according to the method described by Ooi et al. [46] using an MRS medium and incubated at 30 °C for 24 h under anaerobic conditions. As for postbiotic preparation, cell-free supernatant was collected as postbiotics after centrifugation at 10,000× *g* for 15 min at 4°C, whereas for the preparation of paraprobiotics, the cell suspension of *L. plantarum* strains was kept at −30 °C for 7 days to produce paraprobiotics.

### 2.2. Broiler Chicken Management and Experimental Design

A total of 336-day-old COBB 500 chicks (DOCs) were obtained from a commercial hatchery. The DOCs were randomly distributed to eight dietary treatments replicated six times with seven birds per treatment in a completely randomized design (CRD) and managed in a closed house system. The house temperature was set at 33 ± 1 °C on day 1, was gradually reduced to about 25 ± 1 °C by day 15. The average relative humidity ranged between 60 and 75%. Each treatment group was replicated six times with seven birds per replicate and was managed in a 120 × 120 cm (length × width) pen cage. The dietary treatment included T1 = negative control (Basal diet), T2 = positive control (Basal diet + 0.01% (*w*/*w*) Oxytetracycline), T3 = Basal diet + 0.2% (*v*/*w*) postbiotic TL1, T4 = Basal diet + 0.2% (*v*/*w*) postbiotic RS5, T5 = Basal diet + 0.2% (*v*/*w*) paraprobiotic RG11, T6 = Basal diet + 0.2% (*v*/*w*) postbiotic RI11, T7 = Basal diet + 0.2% (*v*/*w*) paraprobiotic RG14, T8 = Basal diet + 0.2% (*v*/*w*) paraprobiotic RI11. The birds were vaccinated against Newcastle disease and infectious bronchitis disease (ND-IB) through eye drops at 7 and 21 days. The infectious bursal disease (IBD) vaccination was done on day 14 by eye drop. Water and feed were offered ad libitum until day 35. The starter and finisher diets (Table 1 and Table 2) were offered from days 0 to 21 and days 22 until 35 of age, respectively. The study was carried out based on the guidelines approved by the Institutional Animal Care and Use Committee of the Universiti Putra Malaysia (IACUC) with reference No: UPM/IACUC/AUP-R098/2018, which ensures that the care and use of animals for scientific purposes is humane and ethical.

### 2.3. Sampling and Data Collection

The measurements for body weight (BW) and feed intake (FI) were done on a weekly basis. The data were used to calculate body weight gain (BWG) and feed conversion ratio (FCR). The European Broiler Index (EBI) was used to compare the broiler results from different treatments. The EBI is a factor that standardizes technical results by considering feed conversion, mortality, and daily gain. EBI was calculated using the formula: (Average grams gained/day × % survival rate)/Feed conversion × 10. At the end of weeks 3 and 5, 6 and 12 chickens were randomly selected from each treatment and slaughtered based on the halal procedure outlined in MS1500:2009 (Department of standard Malaysia, 2009). After being trimmed free of fat and blotted dry with tissue paper, the internal organs were weighed as relative percentages compared to the live body weight. Duodenum, jejunum, and ileum samples of the intestines were taken to determine the crypt depth and villi height of broiler chickens. In contrast, the liver samples were collected to determine IGF-1 and GHR expressions. Blood samples were collected, and plasma was harvested and used for IgA, IgM, and IgG analyses. Colon mucosa scraping was collected for sIgA determination.

### 2.4. Carcass Characteristics and Internal Organs

At the end of week 5, 6 chickens per treatment were randomly picked and slaughtered, de-feathered, and eviscerated for carcass characteristics evaluation. Body parts were cut and weighed individually, including shanks, thighs, drumsticks, wings, back breast, internal organs (liver, heart, gizzard, spleen, and intestines), and abdominal fat. All internal organs and carcass parts were expressed as a percentage of the live body weight using the formula:$$\mathrm{Cut}\;\mathrm{yield}\;\%=\frac{\mathrm{weight}\;\mathrm{of}\;\mathrm{cut}}{\mathrm{Empty}\;\mathrm{body}\;\mathrm{weight}}\times 100$$

### 2.5. Small Intestine Histomorphology

The histomorphology of the small intestine was measured as described in the method used by Choe et al. [31]. After slaughtering the chickens, the intestinal samples were taken for analysis. Measurement for histomorphology was conducted at the Faculty of Veterinary Medicine, Universiti Putra Malaysia. The segment of approximately 5–6 cm long was removed from the ileum (midway between the Meckel’s diverticulum and ileocaecal junction), jejunum (midway between the endpoint of the duodenal loop and Meckel’s diverticulum), and duodenum (the middle part of the duodenal loop). The intestinal sections were flushed with a 10% neutral buffered formalin solution. Afterwards, the segments were excised approximately 3mm and fixed in 10% neutral buffered formalin solution after being transferred into plastic cassettes overnight. The intestinal samples were then dehydrated in a tissue processing machine (Leica ASP 3000, Tokyo, Japan) and embedded in paraffin wax (Leica EG 1160, Tokyo, Japan). The intestinal sections were trimmed at 30 µm, then sections of 4 µm were cut and fixed on the slides in the hot plate at 60°C. The slides were later stained with haematoxylin and eosin, and afterward, were mounted and examined under light microscopes. The depth of the invagination between the adjacent villi (crypt depth) and the height from the tip of the villi to the villi crypt junction (villi height) was measured with an image analyzer.

### 2.6. Immunoglobulins Determination

Six birds were randomly selected and slaughtered by neck decapitation at the end of the starter and finisher phase (weeks 3 and 5 of age). Blood samples were collected into vacutainer tubes containing ethylene diamine tetraacetic acid (EDTA). The tubes with the samples were mixed by gently inverting and were kept temporarily in an ice cube before centrifugation. Plasma was harvested by centrifugation at 3000 rpm for 15 min at 4 °C and kept at −80 °C until further analysis. The plasma samples were used to measure the concentration of immunoglobulin A (IgA), immunoglobulin G (IgG), immunoglobulin M (IgM), and the secretory immunoglobulin A (sIgA) was measured from the colon mucosa.

#### Plasma IgA, IgG, IgM and Colon Mucosa sIgA

The plasma IgA, IgG, IgM, and colon mucosa sIgA were measured using IgG, IgM, and IgA ELISA kits (QAYEE -BIO, Shanghai, China). The standard concentrations were as follows: IgA (300, 150, 75, 37.5, 18.7, 0 µg/mL), IgG (500, 250, 125, 62.5, 31.2, 0 µg/mL), and IgM (100, 50, 25, 12.5, 6.25, 0 µg/mL). Briefly, 50 µL of standard and appropriately diluted sample (in duplicate) were loaded into microplate wells. Horseradish peroxidase (HRP, 50 µL) was added into each well of the standard and sample, sealed and gently shaken, and then incubated for 60 min at 37 °C. After incubation, the well content was discarded and washed five times with a washing solution. Chromogen solutions A and B (50 µL of each) were added to each well and incubated at 37 °C for 10 min in the dark. Immediately after adding 50 µL of stop solution into each well, the absorbance was recorded at 450 nm using a microplate reader (BioTek^TM^ ELx800^TM^, Winooski, VT, USA). A blank containing standard solution (without sample or HRP) was measured, and its absorbance was subtracted from the absorbance of samples and standard. The plasma IgA, IgG, IgM, and colon mucosa sIgA concentrations, were obtained using standard curves for IgA, IgG, IgM, and sIgA.

### 2.7. RNA Isolation and Real Time-PCR Analysis for Hepatic IGF-1 and GHR

The RNA was isolated from liver samples using Macherey Nagel NucleoSpin RNA plus (Macherey- Nagel GmbH and Co. KG, Düren, Germany). Approximately 30mg of liver was mixed with 350 µL of buffer LBP and homogenized. RNA purification and collection were achieved using NucleoSpin gDNA removal column, NucleoSpin RNA plus column, with wash buffer WB1, WB2, and finally, RNase-free water to elute the RNA. The purity and concentration (260/280 nm ratio absorbance) of extracted RNA were measured by a nanodrop 2000 spectrophotometer (Thermo Scientific, Wilmington, DE, USA). Purified RNA was converted to complimentary DNA (cDNA) using Biotechrabbit cDNA kit (Biotechrabbit GmbH Neuendorfstr. 24a 16761 Hennigsdorf, Germany).

Real-time PCR was performed using LightCycler^®^480 Roche (Roche Diagnostics GmbH Roche Applied Science 68298, Roche Diagnostics GmbH Roche, Mannheim, Germany). GAPDH was used as the housekeeping gene to standardize the target genes. The qPCR master mix (20 μL) was made for every sample using a CAPITAL ^TM^ qPCR Green Mix, 4^X^ (Biotechrabbit GmbH Neuendorfstr. 24a 16761 Hennigsdorf, Germany), which comprises 5 µL of CAPITAL ^TM^ qPCR Green Mix, 4×, 4 μM forward and reverse primers of 1 μL of each, 2 μL of template cDNA and 12 μL of Nuclease free water. The targeted gene primers are presented in Table 3.

### 2.8. Statistical Analysis

The experiment followed a completely randomized design (CRD) model. Data generated were analyzed by one-way analysis of variance (ANOVA) using the general linear model (GLM) of the statistical analysis system [47]. Duncan’s Multiple Range Test was used to compare the treatment means at the probability level of 5% (*p* < 0.05).

## 3. Results

### 3.1. Growth Performance of Broilers Fed Postbiotics and Paraprobiotics

The results of the effect of postbiotics and paraprobiotics on the final body weight (FBW), cumulative weight gain (CWG), cumulative feed intake (CFI), and feed conversion ratio (FCR) are shown in Table 4. A significant (*p* < 0.05) change was recorded in the daily CFI across the treatment at the starter phase. However, FBW, CWG, and FCR were not significantly (*p* > 0.05) affected by the dietary treatments. In the finisher phase, the supplementation with postbiotics and paraprobiotics significantly improved the CFI and FCR groups. T8 exhibited the highest CFI, while the lowest FCR was recorded at T6.

The overall growth performance showed no significant difference (*p* > 0.05) in the FBW, CWG, and FCR of chickens across all the groups. However, birds on T1, T5, T7, and T8 recorded higher CFI (*p* < 0.05). The European broiler index (EBI) was calculated and significantly different across the treatment groups. The highest EBI was recorded in T6.

### 3.2. Carcass Characteristics and Internal Organs of Broiler Chicken Fed Postbiotics and Paraprobiotics

The results of carcass characteristics and internal organs are presented in Table 5. The results showed no significant difference (*p* > 0.05) in the carcass weight, carcass yield, breast, drumstick, thigh, wing, back, shank, gizzard, liver, spleen, intestine, and heart across the treatments. However, the abdominal fat was significantly different (*p* < 0.05); T7 had the lowest abdominal fat.

### 3.3. Small Intestine Histomorphology

The results of the villi height, crypt depth, and villi height to crypt depth ratio (VH: CD) of the duodenum, jejunum, and ileum of broilers fed postbiotics and paraprobiotic at 3 and 5 weeks are presented in Table 6. Furthermore, the image of the villi heights and crypt depth of the small intestines as viewed under the light microscope are presented in Figure 1. Birds fed T5 had significantly (*p* < 0.05) higher duodenal villi than the controls and the other treatments. Furthermore, postbiotics and paraprobiotics chickens had higher duodenal villi than the control groups. The T6 chickens had the highest jejunal villi. The dietary treatments did not significantly influence the duodenal crypt depth. However, there was a significant effect in the jejunum crypt depth, and T5 recorded the shortest jejunum crypt depth.

Similarly, a significant difference (*p* < 0.05) was reported in the ileal crypt depth. The shortest crypt depth was recorded at T4. Birds fed T5 and T6 diets had a significantly (*p* < 0.05) higher VH: CD ratio in the duodenum compared to the control groups. T5 demonstrated a significantly (*p* < 0.05) higher VH: CD ratio in the jejunum than the T1. The VH: CD ratio in the ileum was not significantly different.

The birds fed T6 recorded a significantly (*p* < 0.05) higher duodenal villi at the finisher stage, while T8 recorded both the highest jejunal and ileal villi significantly. The duodenal depth was not significantly (*p* > 0.05) impacted by the dietary treatment. The jejunal crypt was significantly affected, and T4 recorded the shortest crypt depth. Similarly, there was a significant (*p* < 0.05) change in the ileum crypt depth. Chickens on T8 diets had a significantly (*p* < 0.05) shorter crypt depth at the ileum. The VH: CD was not significantly different (*p* < 0.05); however, the highest VH: CD was recorded at T6. Jejunal had the highest VH: CD, while the ileum VH: CD was the highest in T8.

### 3.4. Plasma IgA, IgG, IgM and Colon Mucosa sIgA

The plasma and colon mucosa IgA concentrations are presented in Figure 2. No significant difference (*p* > 0.05) was observed in plasma IgA concentrations. Conversely, T5, T6, T7, and T8 recorded significantly higher sIgA in the colon mucosa. The plasma IgG concentrations of the starter and finisher are presented in Figure 3. No significant difference (*p* > 0.05) was recorded at the starter and finisher phases. However, T6 and T8 recorded higher IgG at the starter phase, while T8 recorded a higher value for IgG at the finisher stage. Figure 4 shows the plasma concentrations of IgM for starter and finisher. No significant (*p* > 0.05) changes occurred at the starter stage, but T2 recorded a higher value for IgM. However, T2 had significantly (*p* < 0.05) higher IgM at the finisher stage.

### 3.5. IGF-1 and GHR mRNA Expression

The supplementation of postbiotics and paraprobiotics in broiler chicken diets resulted in a significant upregulation of the expression of GHR and IGF-1 (Figure 5). Birds fed diets supplemented with postbiotics RI11 had a significant (*p* < 0.05) increase in the expression of GHR. Similarly, the mRNA IGF-1 expression was significantly (*p* < 0.05) higher in postbiotic RI11 followed by paraprobiotics RG11.

## 4. Discussion

### 4.1. Growth Performance and Carcass Characteristics

Postbiotics and paraprobiotics showed positive effects on growth performance. The performance of the postbiotics and paraprobiotics chickens was comparable to the positive control (antibiotic-fed chickens) and, in some cases, even better. This could be due to the bacteriostatic and bactericidal properties of postbiotics and paraprobiotics responsible for reducing pathogenic bacteria in the gut. Hence, they can function like antibiotics in terms of enhancing growth performance. Many studies with postbiotics demonstrated the antibacterial property of postbiotics produced from *L. plantarum* [31,32,33,42]. The result from our in vitro study with postbiotics and paraprobiotics showed that paraprobiotics also exhibited antibacterial activity against pathogenic bacteria. The paraprobiotic strains used in this study were from the same *L. plantarum*. Hence, paraprobiotics also naturally contain similar antibacterial compounds, which allow them to inhibit the multiplication of harmful bacteria.

The FBW, CWG, and FCR were comparable to those of both negative and positive controls without significant differences. However, significant (*p* < 0.05) improvements were recorded in FBW and FCR in previous studies with postbiotics [43]. Higher BW and BWG in postbiotics and inulin compared to the negative and positive controls were reported by Kareem et al. [48]. This finding corroborated that postbiotics and paraprobiotics can reduce harmful bacteria and enhance growth, just like antibiotics. Hence, they can be considered good replacements for antibiotics. Recently, probiotics have been receiving wide attention and are emerging as a safe and viable alternative to antibiotics due to their ability to cause improvement in the performance of livestock [49]. Interestingly, however, postbiotics and paraprobiotics are safer than probiotics because they contain functional fermentation compounds such as short-chain fatty acids, microbial fractions, functional proteins, secreted polysaccharides, extracellular polysaccharides, cell lysates, teichoic acid, peptidoglycan-derived muropeptides, and pili-type structure [35,50,51,52,53,54,55], and can be combined with other compounds to improve the health status of animals [56].

Although the overall FCR did not record any significant difference, the birds on TL1 and RI11 had the lowest FCR among the treated groups. This is contrary to the higher FCR reported with postbiotic RI11 [44]. This result could be linked to the controlled environment of the closed house system used in this study, which was different from the opened house system reported by Humam et al. [44].

There was no significant difference in the effects of postbiotics and paraprobiotics on the carcass yield of the broiler chickens. This finding is similar to the one reported by Humam et al. [44], in which feeding postbiotics to broiler chickens under heat stress did not affect their carcass yield. Similarly, carcass yield was not affected when the combination of postbiotics and inulin was fed to broiler chicken [8]. However, this study recorded significant effects on abdominal fat and intestines. The postbiotic and paraprobiotic groups showed a decrease in abdominal fat compared to the negative and positive controls. Birds supplemented with paraprobiotic RG14 recorded the lowest abdominal fat. This was consistent with the report by Loh [9], in which postbiotic metabolites demonstrated potential usefulness in addressing the issues of high meat and egg yolk cholesterol. Furthermore, *L. plantarum* bacteria was reported to cause a reduction of fat deposits in chickens [57].

### 4.2. Small Intestine Histomorphology

The measurement of intestinal morphology can reveal an increase in nutrient absorption when there is increased villus height, short crypt depth, higher villus height-crypt depth ratio, etc. [58]. Furthermore, the height and crypt depth of the villus play an important role in gut function and animal health [59]. Previous findings showed that feeding broilers with postbiotics caused improvement of histomorphology through increased villi height in the duodenum and ileum [60].

Paraprobiotic RG11 and the strain RI11 (both postbiotic and paraprobiotic) contributed to greater improvement in histomorphology than other strains. This finding is in tandem with a recent report by Humam et al. [44]. Enhanced growth performance, high nutrient absorption, and decreased gastrointestinal secretions might be accounted for through increased villi height and decreased crypt depth [61]. According to Jha et al. [58], longer villi indicate feed efficiency and growth-promoting efficiency improvements. Another important finding was the increase in villi heights, VH: CD ratios, and reduced crypt depths than the control groups, similar to those obtained from the study conducted by Humam et al. [44].

Documented evidence revealed that postbiotics could improve histomorphology of the small intestines by increasing beneficial microbes such as LAB, which could cause a decrease in the risk of villi damage caused by gut pathogens [27,28,29,60].

### 4.3. Immune Status

The immune system is composed of an important component called B cells, responsible for producing immunoglobulins [62]. Immunoglobulins play a vital role in immune regulation and mucosal defense, but they are affected by environmental stress factors [44]. IgA is mainly secreted from mucosa membranes and is the most developed immunoglobulin in mammals [63]. IgA plays a crucial role in protecting mucosal surfaces, thus preventing the entry, binding, and colonization of toxins and pathogens [44]. Palm et al. [64] reported that sIgA is the most abundant colonic antigen known as “immune exclusion”. The dietary treatments increased colon mucosa sIgA significantly (*p* < 0.05) in this present work. The sIgA concentrations in the colon mucosa was more in postbiotic RI11, paraprobiotic RI11, and paraprobiotic RG11. Most recently, diets were reported to contribute to the secretion of intestinal sIgA via intestinal microbiota and could be beneficial to the animals’ health [65,66,67], which is consistent with the findings of this study. IgM functions include the regulation of subsequent immune response, facilitating the production of IgG and the first immune response against foreign antigens [68]. In this study, IgM was not affected by the dietary treatment at the starter phase. IgG did not show any significant difference (*p* > 0.05) across the treatment diets at both the starter and finisher phases. In contrast, a significantly (*p* < 0.05) greater level of plasma IgG was reported in chickens fed with postbiotic RI11 under heat stress [44]. The higher dosage of 0.3% postbiotic RI11 fed in their study might have accounted for the significantly greater level of IgG compared to the results of the present study.

### 4.4. IGF1 and GHR mRNA Expression

In poultry, growth hormone (GH) is an important regulator of growth and body composition [69]. GH was responsible for stimulating the hepatic production of IGF-1 through the actions of GH-activated and GH receptors. Many factors, including diet, can influence the expression of hormones such as insulin-growth factor 1 (IGF-1) and growth hormone receptors [70]. Postbiotics and paraprobiotics increased the hepatic expression of GHR and IGF-1. Postbiotic RI11 showed the highest expression of GHR and IGF-1; this agrees with a previous report by Humam et al. [44]. Similarly, Kareem et al. [60] reported upregulation of mRNA GH and IGF-1 in postbiotics and inulin supplementation. Additionally, the supplementation of postbiotics was reported to increase SCFA production with a more abundant intestinal microbiota (e.g., *Lactobacilli* and *Bifidobacterium*) [7]. This could increase IGF-1 expression in the bone marrow, liver, and adipose tissues, as reported in mice supplemented with SCFA [71].

## 5. Conclusions

The feeding of postbiotics and paraprobiotics compared favorably with antibiotics with respect to the growth performance of broiler chicken. Additionally, better performance was recorded for some parameters in some treatments compared to the antibiotics group. The supplementation of postbiotics and paraprobiotics significantly increased feed intake. However, other growth parameters such as FBW, BWG, FCR, and mortality were not significantly affected across the dietary treatment. Carcass yield was not affected, but birds supplemented with paraprobiotic RG14 (T7) had the lowest abdominal fat significantly. Villi heights were increased due to dietary treatments. Colon mucosa sIgA was significantly influenced by the dietary treatments. Similarly, plasma immunoglobulin IgM levels recorded significant changes at the finisher phase. GHR and IGF-1 expressions were upregulated by postbiotics and paraprobiotics supplementation in this current study.

## Figures and Tables

**Figure 1 animals-12-00917-f001:**
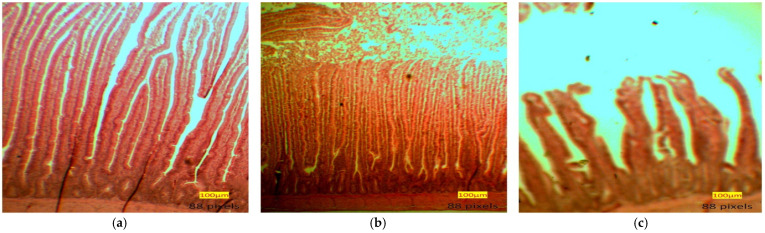
(**a**) Duodenum, (**b**) Jejunum, (**c**) Ileum villi heights and crypt depth of small intestines.

**Figure 2 animals-12-00917-f002:**
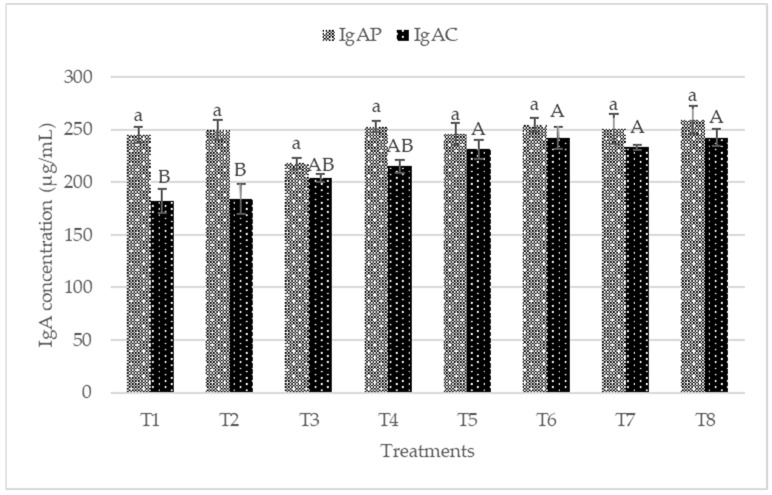
Plasma and Colon mucosa immunoglobulin A (IgA) concentrations of broiler chicken fed postbiotics and paraprobiotics. Bars with the same letters are significantly different (*p* > 0.05) for Plasma IgA, while bars with different letters differs significantly (*p* < 0.05) for Colon mucosa IgA, *n* = 6. T1 = negative control (Basal diet), T2 = positive control (Basal diet + 0.01% (*w*/*w*) Oxytetracycline), T3 = Basal diet + 0.2% (*v*/*w*) postbiotic TL1, T4 = Basal diet + 0.2% (*v*/*w*) postbiotic RS5, T5 = Basal diet + 0.2% (*v*/*w*) paraprobiotic RG11, T6 = Basal diet + 0.2% (*v*/*w*) postbiotic RI11, T7 = Basal diet + 0.2% (*v*/*w*) paraprobiotic RG14, T8 = Basal diet + 0.2% (*v*/*w*) paraprobiotic RI11. IgAP = Plasma immunoglobulin A, IgAC = Colon mucosa immunoglobulin A.

**Figure 3 animals-12-00917-f003:**
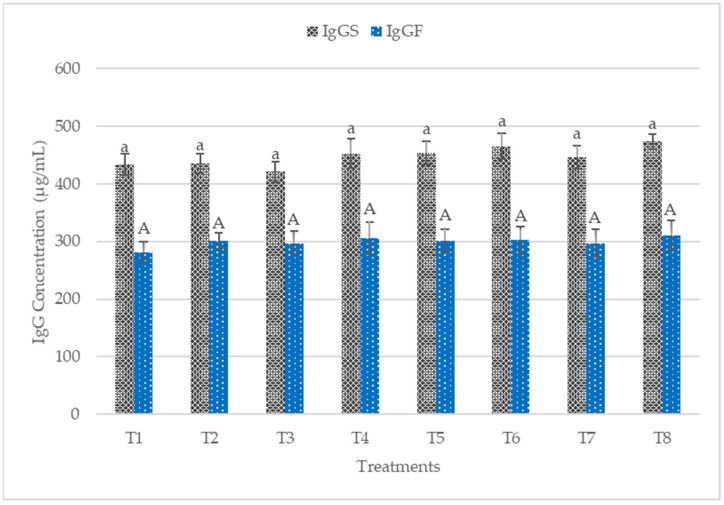
Plasma immunoglobulin G (IgG) concentrations of starter and finisher broiler chicken fed postbiotics and paraprobiotics. Bars with the same letters were not significantly different (*p* > 0.05) for both IgG for starter and IgG for finisher, *n* = 6. T1 = negative control (Basal diet), T2 = positive control (Basal diet + 0.01% (*w*/*w*) Oxytetracycline), T3 = Basal diet + 0.2% (*v*/*w*) postbiotic TL1, T4 = Basal diet + 0.2% (*v*/*w*) postbiotic RS5, T5 = Basal diet + 0.2% (*v*/*w*) paraprobiotic RG11, T6 = Basal diet + 0.2% (*v*/*w*) postbiotic RI11, T7 = Basal diet + 0.2% (*v*/*w*) paraprobiotic RG14, T8 = Basal diet + 0.2% (*v*/*w*) paraprobiotic RI11. IgGS = Immunoglobulin M for starter, IgGF = Immunoglobulin M for finisher.

**Figure 4 animals-12-00917-f004:**
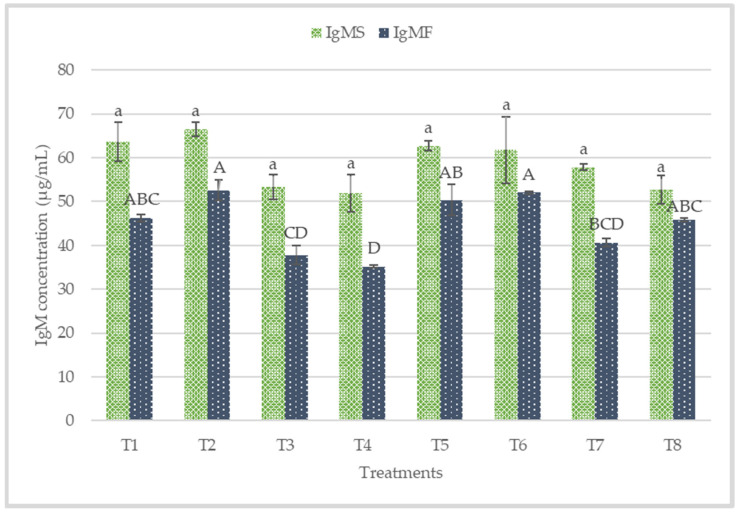
Plasma immunoglobulin concentrations of starter and finisher broiler chicken fed postbiotics and paraprobiotics. Bars with the same letters were not significantly different (*p* > 0.05) for the IgM starter, while Bars with different letters are significantly different (*p* < 0.05) for the IgM finisher, *n* = 6. T1 = negative control (Basal diet), T2 = positive control (Basal diet + 0.01% (*w*/*w*) Oxytetracycline), T3 = Basal diet + 0.2% (*v*/*w*) postbiotic TL1, T4 = Basal diet + 0.2% (*v*/*w*) postbiotic RS5, T5 = Basal diet + 0.2% (*v*/*w*) paraprobiotic RG11, T6 = Basal diet + 0.2% (*v*/*w*) postbiotic RI11, T7 = Basal diet + 0.2% (*v*/*w*) paraprobiotic RG14, T8 = Basal diet + 0.2% (*v*/*w*) paraprobiotic RI11. IgMS = Immunoglobulin M for starter, IgMF = Immunoglobulin M for finisher.

**Figure 5 animals-12-00917-f005:**
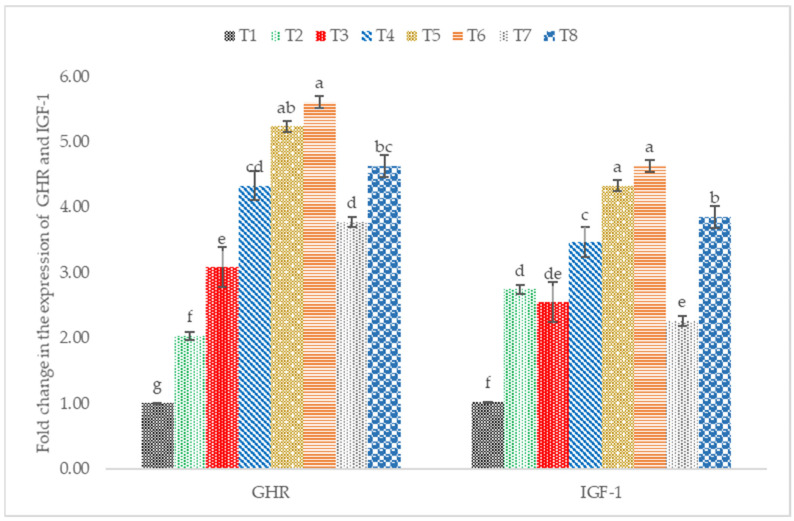
Hepatic GHR and IGF-1 mRNA expression level in broiler chickens fed postbiotics and paraprobiotics. Different letters on standard error bars indicate significant difference (*p* < 0.05) for both GHR and IGF-1, *n* = 6. T1 = negative control (Basal diet), T2 = positive control (Basal diet + 0.01% (*w*/*w*) Oxytetracycline), T3 = Basal diet + 0.2% (*v*/*w*) postbiotic TL1, T4 = Basal diet + 0.2% (*v*/*w*) postbiotic RS5, T5 = Basal diet + 0.2% (*v*/*w*) paraprobiotic RG11, T6 = Basal diet + 0.2% (*v*/*w*) postbiotic RI11, T7 = Basal diet + 0.2% (*v*/*w*) paraprobiotic RG14, T8 = Basal diet + 0.2% (*v*/*w*) paraprobiotic RI11. GHR = Growth Hormone Receptor, IGF-1 = Insulin-like Growth Factor.

**Table 1 animals-12-00917-t001:** Nutrient composition of starter diets (day 1–21).

Ingredients	Treatment diets							
	T_1_	T_2_	T_3_	T_4_	T_5_	T_6_	T_7_	T_8_
Corn	47.50	47.49	47.20	47.20	47.20	47.20	47.20	47.20
Soybean Meal	40.10	40.10	40.20	40.20	40.20	40.20	40.20	40.20
Wheat pollard	1.50	1.50	1.60	1.60	1.60	1.60	1.60	1.60
CPO	6.00	6.00	5.90	5.90	5.90	5.90	5.90	5.90
L-Lysine	0.50	0.50	0.50	0.50	0.50	0.50	0.50	0.50
DL-Methionine	0.50	0.50	0.50	0.50	0.50	0.50	0.50	0.50
Dicalcium Phosphate	2.50	2.50	2.50	2.50	2.50	2.50	2.50	2.50
Calcium carbonate	0.45	0.45	0.45	0.45	0.45	0.45	0.45	0.45
Choline chloride	0.10	0.10	0.10	0.10	0.10	0.10	0.10	0.10
Salt	0.35	0.35	0.35	0.35	0.35	0.35	0.35	0.35
Mineral Mix	0.15	0.15	0.15	0.15	0.15	0.15	0.15	0.15
Vitamin Mix	0.15	0.15	0.15	0.15	0.15	0.15	0.15	0.15
Antioxidant	0.10	0.10	0.10	0.10	0.10	0.10	0.10	0.10
Toxin binder	0.10	0.10	0.10	0.10	0.10	0.10	0.10	0.10
Antibiotics	0.00	0.01	0.00	0.00	0.00	0.00	0.00	0.00
Postbiotic TL1	0.00	0.00	0.20	0.00	0.00	0.00	0.00	0.00
Postbiotic RS5	0.00	0.00	0.00	0.20	0.00	0.00	0.00	0.00
Paraprobiotic RG11	0.00	0.00	0.00	0.00	0.20	0.00	0.00	0.00
Postbiotic RI11	0.00	0.00	0.00	0.00	0.00	0.20	0.00	0.00
Paraprobiotic RG14	0.00	0.00	0.00	0.00	0.00	0.00	0.20	0.00
Paraprobiotic RI11	0.00	0.00	0.00	0.00	0.00	0.00	0.00	0.20
Total	100	100	100	100	100	100	100	100
Calculated analysis								
ME (Kcal/Kg)	3215.70	3215.70	3201.40	3201.40	3201.40	3201.40	3201.40	3201.40
Protein (%)	22.00	22.00	22.03	22.03	22.03	22.03	22.03	22.03
Fat (%)	7.99	7.99	7.88	7.88	7.88	7.88	7.88	7.88
Fibre (%)	4.11	4.11	4.12	4.12	4.12	4.12	4.12	4.12
Calcium (%)	1.08	1.08	1.08	1.08	1.08	1.08	1.08	1.08
Total Phos(%)	0.89	0.89	0.89	0.89	0.89	0.89	0.89	0.89
Avail. P (%)	0.48	0.48	0.48	0.48	0.48	0.48	0.48	0.48

T1 = negative control (Basal diet), T2 = positive control (Basal diet + 0.01% (*w*/*w*) Oxytetracycline), T3 = Basal diet + 0.2% (*v*/*w*) postbiotic TL1, T4 = Basal diet + 0.2% (*v*/*w*) postbiotic RS5, T5 = Basal diet + 0.2% (*v*/*w*) paraprobiotic RG11, T6 = Basal diet + 0.2% (*v*/*w*) postbiotic RI11, T7 = Basal diet + 0.2% (*v*/*w*) paraprobiotic RG14, T8 = Basal diet + 0.2% (*v*/*w*) paraprobiotic RI11. CPO = Crude palm oil. Dicalcium phosphate 18%; Vitamin premix provided per kilogram of diet: Vitamin A 35 MIU; vitamin D3 9 MIU; vitamin E 90 g; vitamin K3 6 g; vitamin B1 7 g; vitamin B2 22 g; vitamin B6 12 g; vitamin B12 0.070 g; biotin 300 mg; pantothenic acid 35 g; nicotinic acid, 120 g; folic acid 3 g; Phytase 25000 FTU. Mineral mix provided per kilogram of diet: Se 0.2 g, Cu 15 g, Fe 80 g, I 1 g, Mn 100 g, Na 1.5 g, Zn 80 g, K 4 g, Co 0.25 g. Antioxidant contains butylated hydroxyanisole (BHA). Toxin binder contains natural hydrated sodium calcium aluminum silicates to reduce the exposure of feed to mycotoxins. Oxytetracycline (200 mg/kg, purity ≥ 64.7%, Y.S.P. Industries (M) SDN BHD). The diets were formulated using FeedLIVE International software (Nonthaburi, Thailand).

**Table 2 animals-12-00917-t002:** Nutrient composition of finisher diets (day 22–35).

Ingredients	Treatment Diets							
	T_1_	T_2_	T_3_	T_4_	T_5_	T_6_	T_7_	T_8_
Corn	51.60	51.59	51.60	51.60	51.60	51.60	51.60	51.60
Soybean Meal	33.50	33.50	33.50	33.50	33.50	33.50	33.50	33.50
Wheat pollard	4.80	4.80	4.60	4.60	4.60	4.60	4.60	4.60
CPO	5.20	5.20	5.20	5.20	5.20	5.20	5.20	5.20
L-Lysine	0.50	0.50	0.50	0.50	0.50	0.50	0.50	0.50
DL-Methionine	0.50	0.50	0.50	0.50	0.50	0.50	0.50	0.50
Dicalcium Phosphate	2.50	2.50	2.50	2.50	2.50	2.50	2.50	2.50
Calcium carbonate	0.45	0.45	0.45	0.45	0.45	0.45	0.45	0.45
Choline chloride	0.10	0.10	0.10	0.10	0.10	0.10	0.10	0.10
Salt	0.35	0.35	0.35	0.35	0.35	0.35	0.35	0.35
Mineral Mix	0.15	0.15	0.15	0.15	0.15	0.15	0.15	0.15
Vitamin Mix	0.15	0.15	0.15	0.15	0.15	0.15	0.15	0.15
Antioxidant	0.10	0.10	0.10	0.10	0.10	0.10	0.10	0.10
Toxin binder	0.10	0.10	0.10	0.10	0.10	0.10	0.10	0.10
Antibiotics	0.00	0.01	0.00	0.00	0.00	0.00	0.00	0.00
Postbiotic TL1	0.00	0.00	0.20	0.00	0.00	0.00	0.00	0.00
Postbiotic RS5	0.00	0.00	0.00	0.20	0.00	0.00	0.00	0.00
Paraprobiotic RG11	0.00	0.00	0.00	0.00	0.20	0.00	0.00	0.00
Postbiotic RI11	0.00	0.00	0.00	0.00	0.00	0.20	0.00	0.00
Paraprobiotic RG14	0.00	0.00	0.00	0.00	0.00	0.00	0.20	0.00
Paraprobiotic RI11	0.00	0.00	0.00	0.00	0.00	0.00	0.00	0.20
Total	100	100	100	100	100	100	100	100
Calculated analysis								
ME (Kcal/Kg)	3180.83	3180.83	3176.68	3176.68	3176.68	3176.68	3176.68	3176.68
Protein (%)	19.92	19.92	19.89	19.89	19.89	19.89	19.89	19.89
Fat (%)	7.29	7.29	7.29	7.29	7.29	7.29	7.29	7.29
Fibre (%)	4.01	4.01	3.99	3.99	3.99	3.99	3.99	3.99
Calcium (%)	1.06	1.06	1.06	1.06	1.06	1.06	1.06	1.06
Total Phos(%)	0.89	0.89	0.88	0.88	0.88	0.88	0.88	0.88
Avail. P (%)	0.48	0.48	0.48	0.48	0.48	0.48	0.48	0.48

T1 = negative control (Basal diet), T2 = positive control (Basal diet + 0.01% (*w*/*w*) Oxytetracycline), T3 = Basal diet + 0.2% (*v*/*w*) postbiotic TL1, T4 = Basal diet + 0.2% (*v*/*w*) postbiotic RS5, T5 = Basal diet + 0.2% (*v*/*w*) paraprobiotic RG11, T6 = Basal diet + 0.2% (*v*/*w*) postbiotic RI11, T7 = Basal diet + 0.2% (*v*/*w*) paraprobiotic RG14, T8 = Basal diet + 0.2% (*v*/*w*) paraprobiotic RI11. CPO = Crude palm oil. Dicalcium phosphate 18%; Vitamin premix provided per kilogram of diet: Vitamin A 35 MIU; vitamin D3 9 MIU; vitamin E 90 g; vitamin K3 6 g; vitamin B1 7 g; vitamin B2 22 g; vitamin B6 12 g; vitamin B12 0.070 g; biotin 300 mg; pantothenic acid 35 g; nicotinic acid, 120 g; folic acid 3 g; Phytase 25000 FTU. Mineral mix provided per kilogram of diet: Se 0.2 g, Cu 15 g, Fe 80 g, I 1 g, Mn 100 g, Na 1.5 g, Zn 80g, K 4 g, Co 0.25 g. Antioxidant contains butylated hydroxyanisole (BHA). Toxin binder contains natural hydrated sodium calcium aluminum silicates to reduce the exposure of feed to mycotoxins. Oxytetracycline (200 mg/kg, purity ≥ 64.7%, Y.S.P. Industries (M) SDN BHD). The diets were formulated using FeedLIVE International software (Nonthaburi, Thailand).

**Table 3 animals-12-00917-t003:** The primer sequences for GHR, IGF-1 and GAPDH genes were used for Real Time-qPCR.

Target Gene	Primer Sequence 5′-3′	Product Size (bp)	Accession No
GHR	F—AACACAGATACCCAACAGCC R—AGAAGTCAGTGTTTGTCAGGG	145	NM_001001293.1
IGF-1	F—CACCTAAATCTGCACGCT R—CTTGTGGATGGCATGATCT	140	NM_001004384.2
GAPDH	F—CTGGCAAAGTCCAAGTGGTG R—AGCACCACCCTTCAGATGAG	275	NM_204305.1

bp = base pair (product size), F = Forward, R-Reverse.

**Table 4 animals-12-00917-t004:** Growth performance of broiler chicken fed postbiotics and paraprobiotics (1–5 weeks).

Parameter	Dietary Treatments									
	T_1_	T_2_	T_3_	T_4_	T_5_	T_6_	T_7_	T_8_	SEM	*p*-Value
				0–3 weeks (starter)						
Initial BW	43.62	43.52	43.14	42.76	43.52	44.52	42.90	43.95	0.54	0.389
FBW (g)	895.00	856.20	847.30	844.20	900.00	1087.20	902.60	886.00	41.63	0.423
CBWG (g)	851.33	812.67	804.15	801.48	856.43	830.81	859.71	842.05	16.58	0.055
CFI (g)	1080.95 ^a^	1073.81 ^a^	999.65 ^c^	1028.91 ^bc^	1088.10 ^a^	1080.95 ^a^	1055.05 ^ab^	1066.32 ^a^	10.23	<0.0001
FCR (g:g)	1.29	1.35	1.27	1.30	1.29	1.32	1.26	1.30	0.03	0.554
				4–5 Weeks (Finisher)						
FBW (g)	2482.39	2419.22	2367.66	2345.17	2420.97	2437.56	2409.2	2471.47	39.61	0.213
CBWG (g)	1563.83	1541.22	1493.03	1487.39	1503.43	1544.33	1490.00	1558.00	29.73	0.332
CFI (g)	2400.00 ^a^	2269.45 ^b^	2234.29 ^b^	2244.45 ^b^	2371.91 ^a^	2272.22 ^b^	2408.57 ^a^	2400.00 ^a^	15.71	<0.0001
FCR (g:g)	1.55 ^abc^	1.49 ^c^	1.52 ^c^	1.55 ^abc^	1.62 ^ab^	1.49 ^c^	1.63 ^a^	1.55 ^abc^	0.33	0.0223
				Overall						
FBW (g)	2482.39	2419.22	2367.66	2345.17	2420.97	2437.56	2409.2	2471.47	39.61	0.213
CBWG (g)	2438.44	2375.44	2324.40	2302.33	2377.66	2392.72	2366.17	2427.18	39.51	0.223
CFI (g)	3480.95 ^a^	3343.25 ^b^	3234.29 ^c^	3273.41 ^c^	3460.88 ^a^	3353.17 ^b^	3464.08 ^a^	3465.59 ^a^	21.55	<0.0001
FCR (g:g)	1.44	1.42	1.41	1.46	1.47	1.41	1.47	1.44	0.03	0.431
Mortality (n)	0/42	0/42	1/42	0/42	1/42	0/42	1/42	2/42		0.517
EBI	483.80 ^b^	478.00 ^c^	459.80 ^d^	450.60 ^g^	451.10 ^f^	484.80 ^a^	448.90 ^h^	458.70 ^e^		<0.0001

Means with different superscripts differs significantly (*p* < 0.05). T1 = negative control (Basal diet), T2 = positive control (Basal diet + 0.01% (*w*/*w*) Oxytetracycline), T3 = Basal diet + 0.2% (*v*/*w*) postbiotic TL1, T4 = Basal diet + 0.2% (*v*/*w*) postbiotic RS5, T5 = Basal diet + 0.2% (*v*/*w*) paraprobiotic RG11, T6 = Basal diet + 0.2% (*v*/*w*) postbiotic RI11, T7 = Basal diet + 0.2% (*v*/*w*) paraprobiotic RG14, T8 = Basal diet + 0.2% (*v*/*w*) paraprobiotic RI11, BW = Body weight, FBW = Final Body weight, CBWG = Cumulative Body weight gain, CFI = cumulative feed intake, FCR = Feed conversion ratio, SEM = Standard error of means, EBI = European Broiler Index.

**Table 5 animals-12-00917-t005:** Carcass weight and carcass yield of broiler chicken fed different strains of *L. plantarum* postbiotic and paraprobiotic.

Parameter	Dietary Treatments									
	T1	T2	T3	T4	T5	T6	T7	T8	SEM	*p*-Value
Carcass weight (g)	1994.50	1834.33	1829.50	1949.83	1824.50	1880.17	1855.33	2019.33	52.04	0.056
Carcass (%)	76.01	74.44	73.75	75.07	73.64	74.09	74.00	76.08	0.91	0.414
Breast (%)	36.84	39.99	38.86	35.55	40.14	40.14	37.89	40.49	1.68	0.643
Drumsticks (%)	11.99	12.80	13.26	12.07	13.14	13.40	12.56	12.32	0.41	0.158
Thigh (%)	14.53	13.59	16.31	13.84	15.44	16.31	14.68	14.45	0.74	0.168
Wings (%)	9.49	10.20	10.37	9.23	9.63	9.72	9.39	9.14	0.35	0.287
Back (%)	23.03	24.06	25.03	22.81	21.96	23.07	22.02	23.14	0.86	0.394
Shanks (%)	4.68	4.93	5.08	4.56	5.18	4.97	5.33	5.09	0.15	0.067
Abdominal fat (%)	1.08 ^a^	1.28 ^ab^	0.93 ^abc^	0.80 ^bc^	0.90 ^abc^	0.89 ^abc^	0.33 ^d^	0.55 ^cd^	0.12	<0.0001
Gizzard (%)	3.14	3.09	3.54	3.49	3.46	3.24	3.28	3.77	0.19	0.316
Liver (%)	3.10	3.10	2.91	2.86	3.51	3.03	3.42	3.08	0.19	0.332
Spleen (%)	0.14	0.22	0.26	0.20	0.15	0.20	0.29	0.21	0.04	0.297
Intestine (%)	4.82 ^a^	5.42 ^a^	5.06 ^a^	4.77 ^a^	4.72 ^a^	4.78 ^a^	3.97 ^b^	4.86 ^a^	0.24	0.032
Heart (%)	0.84	0.94	1.39	0.90	0.83	0.89	0.90	0.89	0.13	0.638

Means with different superscript differs significantly (*p* < 0.05). T1 = negative control (Basal diet), T2 = positive control (Basal diet + 0.01% (*w*/*w*) Oxytetracycline), T3 = Basal diet + 0.2% (*v*/*w*) postbiotic TL1, T4 = Basal diet + 0.2% (*v*/*w*) postbiotic RS5, T5 = Basal diet + 0.2% (*v*/*w*) paraprobiotic RG11, T6 = Basal diet + 0.2% (*v*/*w*) postbiotic RI11, T7 = Basal diet + 0.2% (*v*/*w*) paraprobiotic RG14, T8 = Basal diet + 0.2% (*v*/*w*) paraprobiotic RI11, SEM = Standard error of means.

**Table 6 animals-12-00917-t006:** Histomorphology of small intestine in broiler chicken fed postbiotics and paraprobiotics.

Parameter	Dietary Treatments									
	T1	T2	T3	T4	T5	T6	T7	T8	SEM	*p*-Value
Villi height Wk 3, µm										
Duodenal	604.30 ^c^	605.4 ^c^	814.10 ^abc^	977.60 ^ab^	1174.10 ^a^	1064.6 ^ab^	791.70 ^bc^	857 ^abc^	78.58	0.0212
Jejunal	691.52 ^a^	518.30 ^b^	664.41 ^a^	714.08 ^a^	779.16 ^a^	812.56 ^a^	745.01 ^a^	759.27 ^a^	43.42	0.0146
Ileal	549.4 ^ab^	575.21 ^a^	382.37 ^cd^	313.23 ^d^	580.98 ^a^	530.4 ^abc^	403.1 ^bcd^	601.81 ^a^	47.74	0.0050
Crypt depth Wk 3, µm										
Duodenal	153.36	139.26	149.13	126.34	122.83	125.37	159.87	127.21	10.85	0.4196
Jejunal	123.91 ^abc^	110.55 ^c^	115.16 ^bc^	152.04 ^a^	101.07 ^c^	143.22 ^ab^	141.51 ^ab^	147.75 ^a^	8.13	0.0080
Ileal	99.49 ^ab^	101.77 ^ab^	105.22 ^a^	73.97^c^	100.48 ^ab^	98.54 ^ab^	78.50 ^ac^	121.15 ^a^	6.70	0.0117
Villi height: Crypt depth										
Duodenal	4.01 ^c^	4.97 ^bc^	5.50 ^c^	7.75 ^ab^	9.68 ^a^	8.63 ^a^	5.00 ^bc^	6.705 ^ab^	0.84	0.0057
Jejunal	5.60 ^b^	4.77 ^b^	5.77 ^b^	4.70 ^b^	7.70 ^a^	5.68 ^b^	5.27 ^b^	5.30 ^b^	0.33	0.0025
Ileal	5.53	5.71	3.75	4.24	5.78	5.36	5.29	5.15	0.44	0.0908
Villi height Wk 5, µm										
Duodenal	906.86 ^b^	944.93 ^b^	905.21 ^b^	854.12 ^b^	813.36 ^b^	1145.04 ^a^	1116.48 ^a^	948.74 ^b^	43.51	0.0008
Jejunal	823.96 ^bc^	547.32 ^d^	848.86 ^bc^	787.94 ^c^	872.22 ^bc^	928.32 ^ab^	760.69 ^c^	1036.25 ^a^	30.42	<0.0001
Ileal	554.61 ^b^	645.09 ^b^	618.35 ^b^	692.09 ^b^	648.04 ^b^	717.18 ^b^	622.60 ^b^	927.37 ^a^	50.51	0.0162
Crypt depth Wk 5, µm										
Duodenal	153.83	142.25	131.36	139.10	110.67	137.76	132.21	134.55	11.50	0.5842
Jejunal	164.12 ^ab^	172.96 ^a^	132.51 ^bcd^	112.33 ^d^	122.02 ^cd^	135.01 ^bcd^	141.63 ^abcd^	154.09 ^abc^	9.94	0.0212
Ileal	154.31 ^a^	143.34 ^ab^	130.40 ^abc^	125.92 ^abc^	118.07 ^bc^	138.09 ^ab^	132.30 ^abc^	107.00 ^c^	8.80	0.0501
Villi height: Crypt depth										
Duodenal	6.22	6.66	6.96	6.20	7.39	8.95	8.43	7.20	0.77	0.3957
Jejunal	5.13 ^c^	3.18 ^d^	6.41 ^abc^	7.05 ^ab^	7.17 ^a^	6.92 ^ab^	5.49 ^bc^	6.87 ^ab^	0.43	0.0003
Ileal	3.62b	4.52 ^b^	4.79 ^b^	5.56 ^b^	5.42 ^b^	5.22 ^b^	4.66 ^b^	8.85 ^a^	0.48	0.0006

Means with different superscripts differs significantly (*p* < 0.05). T1 = negative control (Basal diet), T2 = positive control (Basal diet + 0.01% (*w*/*w*) Oxytetracycline), T3 = Basal diet + 0.2% (*v*/*w*) postbiotic TL1, T4 = Basal diet + 0.2% (*v*/*w*) postbiotic RS5, T5 = Basal diet + 0.2% (*v*/*w*) paraprobiotic RG11, T6 = Basal diet +0.2% (*v*/*w*) postbiotic RI11, T7 = Basal diet + 0.2% (*v*/*w*) paraprobiotic RG14, T8 = Basal diet + 0.2% (*v*/*w*) paraprobiotic RI11, SEM = Standard error of means.

## Data Availability

Not Applicable.

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
