# Peer review of "Effects of Postbiotics and Paraprobiotics as Replacements for Antibiotics on Growth Performance, Carcass Characteristics, Small Intestine Histomorphology, Immune Status and Hepatic Growth Gene Expression in Broiler Chickens"

_animals, 2022, doi:10.3390/ani12070917_

Round 1
Reviewer 1 Report
This study investigated supplementation of postbiotics and parabiotics of different strains of L. plantarum in diets and its effects on broilers. Overall, the paper was well written and organized. The major problem is no sufficient information of post- or para-biotics that can provide readers or field valuable, for example strains name, composition of postbiotics. The other major problem is statistics. If the treatments of postbiotics compared to the ones of parabiotics may give more information to see their differences? All results and discussion in the present form are mixed but they may have separated mechanisms on functions.
Minor:
- Treatment names can be re-named as NC (negative control), PC (positive control), T-post-1, T-post-2..... T-para-1...... In the current form, T3,4,6 are postbiotics and T5,7,8 are parabiotics treated, which is randomly named and hard to be recoganized.
- Figures 2-5, legends are not completely described. I believed that, for example, immunoglobulins of plasma and colon are not statistically calculated together but separately statistically analyzed, but legends did not say. Moreover, n should be addressed in legends in each figure with variation bars.
- Table, CPO is not defined.
- Lines 426 - 430, needs reference.
Author Response
|
1 |
Treatment names can be re-named as NC (negative control), PC (positive control), T-post-1, T-post-2.... T-para-1...... In the current form, T3,4,6 are postbiotics and T5,7,8 are parabiotics treated, which is randomly named and hard to be recoganized
|
Thank you for your suggestion. However, the names have been clearly stated and explained as T1=negative control (Basal diet), T2= positive control (Basal diet + 0.01% (w/w)Oxytetracycline), T3=Basal diet + 0.2% (v/w) postbiotic TL1, T4=Basal diet + 0.2% (v/w) postbiotic RS5, T5= Basal diet + 0.2% (v/w) paraprobiotic RG11, T6:=Basal diet + 0.2% (v/w) postbiotic RI11, T7=Basal diet + 0.2% (v/w) paraprobiotic RG14, T8=Basal diet + 0.2% (v/w) paraprobiotic RI11, Lines 38-41. Hence, the names are clearly identified not random in our opinion |
|
2 |
Figures 2-5, legends are not completely described. I believed that, for example, immunoglobulins of plasma and colon are not statistically calculated together but separately statistically analyzed, but legends did not say. Moreover, n should be addressed in legends in each figure with variation bars. |
The necessary additions to the legends have been done and captured as follows:
Figure 2. Plasma and Colon mucosa immunoglobulin A (IgA) concentrations of broiler chicken fed postbiotics and paraprobiotics. Bars with the same letters are significantly different (p>0.05) for Plasma IgA, while bars with different letters differs significantly (p<0.05) for Colon mucosa IgA, n=6. T1 = negative control (Basal diet), T2 = positive control (Basal diet + 0.01% (w/w) Oxytetracycline), T3 = Basal diet + 0.2% (v/w) postbiotic TL1, T4 = Basal diet + 0.2% (v/w) postbiotic RS5, T5 = Basal diet+ 0.2% (v/w) paraprobiotic RG11, T6 = Basal diet + 0.2% (v/w) postbiotic RI11, T7 = Basal diet + 0.2% (v/w) paraprobiotic RG14, T8 = Basal diet + 0.2% (v/w) paraprobiotic RI11. IgAP = Plasma immunoglobulin A, IgAC = Colon mucosa immunoglobulin A, Lines 379-386
Figure 3. Plasma immunoglobulin G (IgG) concentrations of starter and finisher broiler chicken fed postbiotics and paraprobiotics. Bars with the same letters were not significantly different (p>0.05) for both IgG for starter and IgG for finisher, n=6. T1= negative control (Basal diet), T2= positive control(Basal diet + 0.01% (w/w) Oxytetracycline), T3=Basal diet + 0.2% (v/w) postbiotic TL1, T4=Basal diet + 0.2% (v/w) postbiotic RS5, T5= Basal diet+ 0.2% (v/w) paraprobiotic RG11, T6=Basal diet + 0.2% (v/w) postbiotic RI11, T7=Basal diet + 0.2% (v/w) paraprobiotic RG14, T8=Basal diet + 0.2% (v/w) paraprobiotic RI11. IgGS= Immunoglobulin M for starter, IgGF= Immunoglobulin M for finisher, Line 388- 395.
Figure 4. Plasma immunoglobulin concentrations of starter and finisher broiler chicken fed postbiotics and paraprobiotics. Bars with the same letters were not significantly different (p>0.05) for the IgM starter, while Bars with different letters are significantly different (p<0.05) for the IgM finisher, n=6. T1 = negative control (Basal diet), T2 = positive control (Basal diet + 0.01% (w/w) Oxytetracycline), T3 = Basal diet + 0.2% (v/w) postbiotic TL1, T4 = Basal diet+ 0.2% (v/w) postbiotic RS5, T5 = Basal diet + 0.2% (v/w) paraprobiotic RG11, T6 = Basal diet + 0.2% (v/w) postbiotic RI11, T7 = Basal diet + 0.2% (v/w) paraprobiotic RG14, T8 = Basal diet + 0.2% (v/w) paraprobiotic RI11. IgMS = Immunoglobulin M for starter, IgMF = Immunoglobulin M for finisher, Lines 398-405.
Figure 5. Hepatic GHR and IGF-1 mRNA expression level in broiler chickens fed postbiotics and paraprobiotics. Different letters on standard error bars indicate significant difference (p<0.05) for both GHR and IGF-1, n=6. T1 = negative control (Basal diet), T2 = positive control (Basal diet + 0.01% (w/w) Oxytetracycline), T3 = Basal diet + 0.2% (v/w) postbiotic TL1, T4 = Basal diet + 0.2% (v/w) postbiotic RS5, T5 = Basal diet + 0.2% (v/w) paraprobiotic RG11, T6 = Basal diet + 0.2% (v/w) postbiotic RI11, T7 = Basal diet + 0.2% (v/w) paraprobiotic RG14, T8 = Basal diet + 0.2% (v/w) paraprobiotic RI11. GHR = Growth Hormone Receptor, IGF-1 = Insulin-like Growth Factor, Lines 414-420. |
|
3 |
Table, CPO is not defined. |
CPO has been defined as CPO=Crude palm oil and captured accordingly, Lines 184-185 and 198-199. |
|
4 |
Lines 426 - 430, needs reference |
The statement was referred to our invitro studies which have not been published yet. But the manuscript for the invitro studies have been prepared and is awaiting submission. |
Reviewer 2 Report
Scientific article well prepared in terms of content. He adds more knowledge and new sources about alternatives to feed antibiotics. I recommend the article to be published in the journal Animals.

Author Response
|
1 |
Scientific article well prepared in terms of content. He adds more knowledge and new sources about alternatives to feed antibiotics. I recommend the article to be published in the journal Animals |
Thanks for your commendation on our article |
Reviewer 3 Report
Comments to the Authors of manuscript number: animals- entitled “Effects of Postbiotics and Paraprobiotics as Replacements for Antibiotics on Growth Performance, Carcass Characteristics, Small Intestine Histomorphology, Immune Status and Hepatic Growth Gene Expression in Broiler Chickens”.
- L 75 – postbiotics are organic acids, bacteriocins, carbonic substances and enzymes, and it should be explained exactly
- L 147 – small letter
- L 157 – if end products of bacterial metabolism is considered postbiotics and have a therapeutic benefit, what was detected in the supernatant. It should be defined.
- L147, 165-168 different strains of L. plantarum were used in this study. Why these strains were used in different ways? RG11, RG14 and RI11 were used to prepare paraprobiotics; and RI11, RS5, and TL1 were used to prepare postbiotics. It should be explained. There is no information about their properties, if their effect is similar.
- Why these postbiotics and paraprobiotics were introduced at the same concentration? The effect of inactive microbial cells although considered a health benefit is not the same as organic acids produced by bacteria as their metabolic by-product. It is not explained and neither hypotisized.
- l 206 – how the feed intake was measured? For one cage or for one bird? Please define the experimental unit in manuscript.
- L 215 –what part of each mentioned segment? How it was sampled? It should be described.
- L 218 – when blood was collected and how?
- L 218 – what is harvesting of blood plasma?
- L 211 – it is not clear when and how many birds were chosen.
- L 222 – 35 days it is 5 weeks. Uniform the description along the text
- L 231- repetition
- L 244 –small letter
- L 246 – what image analyzer? It should be added
- Figure 1 – it is not at the scientific value.
Author Response
|
1 |
L 75 – postbiotics are organic acids, bacteriocins, carbonic substances and enzymes, and it should be explained exactly |
In L 75, postbiotics was defined as any factor resulting from the activity of a probiotic or any released molecule capable of conferring beneficial effects to the host directly or indirectly [9], Line 75. According to our previous studies, postbiotics were found to contain organic acids, bacteriocins, pyrrole and cyclic compounds (Kareem et al.,2014; Chang et al.,2021). Therefore, those substances mentioned can be found in postbiotics but they cannot be referred to as postbiotics. |
|
2 |
L 147 – small letter |
The capital letter ‘P’ in postbiotics have been corrected to start with small letter, Line 149. |
|
3 |
157 – if end products of bacterial metabolism is considered postbiotics and have a therapeutic benefit, what was detected in the supernatant. It should be defined. |
The supernatant (postbiotics) contain several compounds which made them to possessed therapeutic benefits as previously reported by our research team (Kareem et al.,2014; Chang et al.,2021). However, L157 was just describing the how postbiotics was prepared. |
|
4 |
L147, 165-168 different strains of L. plantarum were used in this study. Why these strains were used in different ways? RG11, RG14 and RI11 were used to prepare paraprobiotics; and RI11, RS5, and TL1 were used to prepare postbiotics. It should be explained. There is no information about their properties, if their effect is similar. |
These strains were selected and used for the feeding trial based on the highest pool of their inhibitory and enzyme activities which were tested in our in vitro studies but yet to be published. |
|
5 |
Why these postbiotics and paraprobiotics were introduced at the same concentration? The effect of inactive microbial cells although considered a health benefit is not the same as organic acids produced by bacteria as their metabolic by-product. It is not explained and neither hypotisized. |
The concentration used for both postbiotics and paraprobiotics was the same because it represents the cumulative concentration of inhibitory and enzyme activity for each of the strains used in the in vitro studies. That was why 0.2% was used for both postbiotics and pararobiotics. |
|
6 |
l 206 – how the feed intake was measured? For one cage or for one bird? Please define the experimental unit in manuscript. |
Feed intake was measured in grams (g) by weighing the leftover feed and subtracting it from the weight of feed offered. It was done per cage and divided by the number of birds in the cage to get the feed intake per bird. The feed intake was done on weekly basis, Lines 209-210. |
|
7 |
L 215 –what part of each mentioned segment? How it was sampled? It should be described. |
The part of each segment has been described in section 2.5: The segment of approximately 5-6 cm long was removed from the ileum (midway between the Meckel’s diverticulum and ileo-caecal junction), jejunum (midway between the endpoint of the duodenal loop and Meckel’s diverticulum), and duodenum (the middle part of the duodenal loop), Line 236-239. |
|
8 |
L 218 – when blood was collected and how? |
When was the collection of the blood samples was done and why was captured in section 2.6: At the end of the starter and finisher phase (weeks 3 and 5 of age), six birds were randomly selected and slaughtered by neck decapitation? Blood samples were collected into vacutainer tubes containing ethylene diamine tetraacetic acid (EDTA), Line 252-253. |
|
9 |
L 218 – what is harvesting of blood plasma? |
Harvesting of blood plasma is a description of how the plasma was collected from the blood after centrifugation as captured in section 2.6: The tubes with the samples were mixed by gently inverting and were kept temporarily in an ice cube before centrifugation. Plasma was harvested by centrifugation at 3000 rpm for 15 min at 4°C and kept at -80°C until further analysis, Line 253-256. |
|
10 |
211 – it is not clear when and how many birds were chosen |
Six birds were chosen at the end of week for starter sampling, whereas 12 birds were chosen for finisher sampling as captured in section 2.3: At the end of weeks 3 and 5, six and twelve chickens were randomly selected from each treatment and slaughtered based on the Halal procedure outlined in MS1500:2009 (Department of standard Malaysia, 2009, Line 214-217. |
|
11 |
L 222 – 35 days it is 5 weeks. Uniform the description along the text |
Corrected as: at the end of weeks 5, Line 225. |
|
12 |
L 231- repetition |
weeks 3 and 5 have been removed to avoid the repetition, Line 234. |
|
13 |
L 244 –small letter |
crypt has been corrected small letter. Line 315 |
|
14 |
L 246 – what image analyzer? It should be added |
The image analyzer is a software that comes with a digital camera which is attached to the microscope to help in measuring the villi heights and the crypt depth. This was how it’s been reported previously in many of our published articles (Choe et l., 2013) |
|
15 |
Figure 1 – it is not at the scientific value |
Figure 1 may not have any scientific value, however, it conveys a proof of the histomorphology analyses that was carried out in the study. |
Reviewer 4 Report
This study investigated the effects of replacing antibiotics with postbiotics 17 and paraprobiotics on growth performance, small intestine morphology, immune status and hepatic 18 growth gene expression in broiler chickens. The study is ethically acceptable and contains sufficient novel data to justify publication in Animals. Therefore, my decision is “accepted” for publication in Animals.
Author Response
|
This study investigated the effects of replacing antibiotics with postbiotics 17 and paraprobiotics on growth performance, small intestine morphology, immune status and hepatic 18 growth gene expression in broiler chickens. The study is ethically acceptable and contains sufficient novel data to justify publication in Animals. Therefore, my decision is “accepted” for publication in Animals. |
Your comments are highly appreciated. Thank you. |
Round 2
Reviewer 1 Report
Figures 2,3,4 should differently label indicating significant differences for the indexes of Starter Phase (eg. a,b,c) and Finisher Phase (eg, A,B,C).
Author Response
Figures 2,3,4 should differently label indicating significant differences for the indexes of Starter Phase (eg. a,b,c) and Finisher Phase (eg, A,B,C).
Thank you for your suggestion. It has been amended as suggested. Please refer to the attached file.
